# Transcription Profiles Reveal the Regulatory Synthesis of Phenols during the Development of Lotus Rhizome (*Nelumbo nucifera* Gaertn)

**DOI:** 10.3390/ijms20112735

**Published:** 2019-06-04

**Authors:** Ting Min, Yinqiu Bao, Baixue Zhou, Yang Yi, Limei Wang, Wenfu Hou, Youwei Ai, Hongxun Wang

**Affiliations:** 1College of Food Science and Engineering, Wuhan Polytechnic University, Wuhan 430023, China; minting1323@163.com (T.M.); baoyinqiu@163.com (Y.B.); zhoubaixue25@163.com (B.Z.); yiy86@whpu.edu.cn (Y.Y.); hwf407@163.com (W.H.); aywlingyun@126.com (Y.A.); 2School Biology and Pharmaceutical Engineering, Wuhan Polytechnic University, Wuhan 430023, China; wanglimeiyx@163.com

**Keywords:** RNA-seq, lotus rhizome, phenols, qRT-PCR

## Abstract

Lotus (*Nelumbo nucifera* Gaertn) is a wetland vegetable famous for its nutritional and medicinal value. Phenolic compounds are secondary metabolites that play important roles in the browning of fresh-cut fruits and vegetables, and chemical constituents are extracted from lotus for medicine due to their high antioxidant activity. Studies have explored in depth the changes in phenolic compounds during browning, while little is known about their synthesis during the formation of lotus rhizome. In this study, transcriptomic analyses of six samples were performed during lotus rhizome formation using a high-throughput tag sequencing technique. About 23 million high-quality reads were generated, and 92.14% of the data was mapped to the reference genome. The samples were divided into two stages, and we identified 23,475 genes in total, 689 of which were involved in the biosynthesis of secondary metabolites. A complex genetic crosstalk-regulated network involved in the biosynthesis of phenolic compounds was found during the development of lotus rhizome, and 25 genes in the phenylpropanoid biosynthesis pathway, 18 genes in the pentose phosphate pathway, and 30 genes in the flavonoid biosynthesis pathway were highly expressed. The expression patterns of key enzymes assigned to the synthesis of phenolic compounds were analyzed. Moreover, several differentially expressed genes required for phenolic compound biosynthesis detected by comparative transcriptomic analysis were verified through qRT-PCR. This work lays a foundation for future studies on the molecular mechanisms of phenolic compound biosynthesis during rhizome formation.

## 1. Introduction 

Lotus (*Nelumbo nucifera* Gaertn), a popular aquatic vegetable in Asian countries [1], is famous for its nutritional and medicinal value [2,3]. Lotus belongs to the eudicot family of *Nelumbonaceae* and is divided into two species, *Nelumbo nucifera* Gaertn. (Asian lotus or sacred lotus) and *Nelumbo lutea* Pers. (American lotus) [4]. Sacred lotus has been cultivated in China for over 2000 years and has gradually developed into two ecotypes due to its different biological properties (rhizome, flowers and seeds) [5]; the temperate type produces an enlarged rhizome and the tropical type has a long period of florescence. 

Rhizomes are produced by modified stems which also can be used for the asexual propagation of lotus [6]. The rhizome of lotus root is formed in the submerged environment; after sprouting, rhizomes start enlarging in one direction and form the floating leaves from the nodes. When the rhizomes grow longitudinally to about 10–20 cm, they begin to swell with carbohydrate storage, which helps the lotus survive over its following life cycle [7,8]. The development stage of rhizomes can be classified as follows: the stolon stage, initial swelling stage, middle swelling stage, and later swelling stage. Rhizome formation is a complex process that depends on the photoperiod and temperature, and the molecular mechanisms of rhizome formation are largely unknown [9]. 

Lotus rhizome (often termed lotus root) is an important vegetable rich in starch, protein, vitamins and mineral substances, which are part of the daily diet [10]. However, the products of lotus rhizome easily become brown, threatening their nutrients and commercial value. Browning has become one of the most important limitations in the storage and processing of lotus rhizome. The reasons for browning are divided into two types: enzymatic and nonenzymatic browning. Enzymatic browning is mostly a result of catalyzing phenols into o-quinones by polyphenol oxidase (i.e., polyphenol oxidase, peroxidase, o-diphenol oxidase and catechol oxidase) [11]. Nonenzymatic browning results from a chemical process called the Maillard reaction. Another aspect is that the phenolic substances provide the lotus rhizome with anti-oxidation, anti-mutagen, anti-aging, and anti-inflammatory properties and pathogen resistance. Phenols are also an important source for traditional medicine [12]. Previous studies have shown that polyphenol extract from banana peel prevents mice from hepatic injury [13]. Tea polyphenols protect skin from photoaging [14]. Polyphenolic extracts of lotus rhizome alleviate hepatic steatosis in mice [15]. Phenolic compounds are also involved in the defense against herbivores, antibacterial and antifungal activity, fruit coloring, pollination and other plant processes. Therefore, research into the biosynthesis and transcriptome metabolism of phenols during rhizome development is urgently needed. 

Phenols are a variety of important and complex secondary metabolites in plants including colored anthocyanins, flavonoids, flavonols, phenolic acids, isoflavones, etc. Phenolic compounds contain benzene rings with one or more hydroxyl substituents and form highly polymerized compounds, largely enriching the variety. Phenolic compounds are synthesized mainly through the phenylalanine metabolic pathways in dicotyledons [16]. First, glucose-6-phosphate is irreversibly transformed to ribulose-5-phosphate through the pentose phosphate pathway (PPP) by glucose-6-phosphate dehydrogenase (G6PDH); this reaction consumes nicotinamide adenine dinucleotide (NAD) and generate nicotinamide adenine dinucleotide phosphate (NADPH) through the catalysis of G-6-P dehydrogenase and 6-phosphogluconolactone dehydrogenase. Ribulose-5-phosphate is the precursor of erythrose-4-phosphate, a substrate for the shikimate pathway. Then, erythrose-4-phosphate reacts with phosphoenolpyruvate (PEP) by glycolysis for further reaction through the shikimate pathway, which converts sugar phosphates to aromatic amino acids (shikimate–chorismate–cinnamate–phenolics). This pathway is also involved in the synthesis of other secondary metabolites (e.g., lignins, salicylates, flavonoid phytoalexins and pigments) that play a critical role in plant growth, development, defense and metabolism. Finally, these phenolics are generated as phenolic compounds through the phenylpropanoid pathway [17].

High-throughput sequencing technology has been used to map and delve into detailed data of the genomes, transcriptomes, proteomes, and metabolomes of living organisms, tissues, organelles, and even single cells [18,19,20]. In recent years, RNA sequencing (RNA-seq) technology has attracted more attention in analyzing the growth and development of plants in terms of maturity and aging, especially fruits and vegetables. Zenoni discovered that grapes undergo two stages of development. A total of 17,324 genes were found throughout development, and 6695 specific genes were expressed during grape maturation [21]. In this study, the tuber part of the lotus rhizome was used as the object to explore the accumulation of phenolic substances at different growth and development stages (S1, S2, S3, S4, S5, S6). The specific genes involved in phenolic anabolic pathways during the growth and development of the lotus rhizome were analyzed. The mechanisms of biosynthesis of lotus phenols were preliminarily elucidated in order to provide a theoretical basis to understand the development of functional activity of lotus phenolics for post-harvest storage and processing.

## 2. Results

### 2.1. Transcriptome of Lotus Rhizome 

Six samples were collected during rhizome development. With the enlargement and swelling of this storage organ, the length was gradually increased from 10 to 25 cm (Figure 1A). A total of 18 samples were tested using the BGISEQ-500 platform. Using RNA-seq technology, we sequenced six libraries, with three biological replicates for each sample. In total, the six libraries generated 21.94–28.4 million raw reads. After removing reads containing adapters or poly-N and low-quality reads, the total number of clean reads per library was in the range of 21.75–25.6 million. We used hierarchical indexing for spliced alignment of transcripts (HISAT) to compare clean reads to the reference genome, “China Antique” reference genome 2 (http://www.ncbi.nlm.nih.gov/genome/genomes/14095), using Bowite2. The comparison results were counted as shown in Table 1. Among the short clean reads, 81.09–96.62% were aligned against the “China Antique” reference genome 2, and 64.29–77.37% of the clean reads were uniquely aligned against the reference genome. The phenolic profile data of different samples can be seen in Appendix A.

Principal component analysis (PCA) is a multivariate statistical analysis method that reduces multiple dimensions to a few independent variables (i.e., principal components) while preserving as much of the original data as possible. In transcriptome analysis, PCA reduces the large amount of gene expression information contained in the sample into a few independent principal components for comparison between samples, which is convenient for finding outlier samples and discriminating samples with high similarity, clusters, etc. The PCA analysis results of this project are shown in Figure 1B. Two samples (S1, S2) formed a single clade, while the other four samples (S3, S4, S5, S6) were clustered into one group. The results show that there was a strong expression differentiation between S3 and S2. In order to reflect the correlation of gene expression between samples, the Pearson correlation coefficient of all gene expression levels between each sample was calculated, and these coefficients were reflected in the form of a heat map, shown in Figure 1C. The correlation coefficient reflects the overall gene expression between samples; the higher the correlation coefficient, the more similar the gene expression level. The x and y axes each represent samples. The color represents the correlation coefficient (the darker the color, the higher the correlation; the lighter the color, the lower the correlation). The results show that samples S1 and S2 were more similar in gene expression and analysis, and that S3, S4, S5, and S6 were more similar in gene expression. This result is corroborated by the PCA analysis (Figure 1B). 

We also performed a distribution of gene expression levels in each sample (Figure 1D), and the degree of dispersion of the data distribution can be observed. The x-axis shows the sample name and the y-axis shows log10 (FPKM (Fragments Per Kilobase of exon model per Million mapped reads) + 1). The box plot for each region corresponds to five statistics (upper to lower, upper quartile, median, and lower quartile, lower limit, where the upper and lower limits do not take outliers into account). Gene expression levels in S1 and S2 are intensive, while in other samples, the expression levels are higher and unconsolidated.

### 2.2. Genes Expression Pattern during Rhizome Formation

Some genes have similar expression patterns at different time points. According to the gene expression information, they can be clustered into time-associated gene clusters. Mfuzz time series analysis software was used based on the loose clustering algorithm. Mfuzz, a software package based on R, can cluster genes according to their similar expression profiles and help to find genes with similar functions. Genes with consistent expression may participate in the same biological process and help in discovering regulatory genes. Genes with the same pattern of expression will be clustered into the same cluster. This method analyzes tissue specificity between samples. Figure 2 shows the case clustering of genes. As shown in Figure 2, 12 clusters were plotted with expression patterns; cluster 1 shows the downregulation of 2856 genes in rhizome formation; cluster 2 shows the upregulation of 2108 genes in the developmental stage; clusters 1, 6, and 7 are significantly expressed in S2; and cluster 3 is significantly expressed in S5. Genes in cluster 8 are highly expressed in S3 and S4, while cluster 9 showed the opposite expression in S3 and S4 from cluster 8. 

### 2.3. Correlation between Transcriptomes of Different Stages of Rhizome Development

To obtain a comprehensive understanding of genes expressed in successive developmental stages and to identify the specific genes that are highly associated with rhizome formation, weighted gene co-expression network analysis (WGCNA) was performed [22]. After filtering out the genes with a low expression (FPKM < 0.05), 1303 genes were retained for WGCNA. Co-expression networks were constructed on the basis of pairwise correlations of gene expression across all samples. Modules were defined as clusters of highly interconnected genes, and genes within the same cluster had high correlation coefficients between them. This analysis identified 30 distinct modules, shown in the dendrogram in Figure 3A (labeled with different colors), in which two major branches define the modules. The 30 modules were correlated with distinct samples due to sample-specific expression profiles. Notably, most modules are composed of genes that were highly expressed in the S1 and S2 sample types; these are indicated in red in Figure 3B.

### 2.4. Metabolic Pathway Analysis of Candidate Genes Involved in Phenols Synthesis 

Gene ontology (GO) is divided into three major functional categories: molecular functions, cellular components and biological processes. Functional classification based on differential gene detection results. There are sub-categories of each level under each major category. As shown in Figure 4A, we further analyzed the enriched GO terms of biological processes; 23,475 genes were annotated by GO assignments and categorized into three major groups (cellular component, molecular function, and biological process) and 48 sub-categories: 5630 of those annotated belonged to the metabolic process group and 409 to the development process group. 

To further explore the biological pathways involved in the expressed genes, we performed a general Kyoto Encyclopedia of Genes and Genomes (KEGG) analysis of genes. The KEGG metabolic pathway of genes is divided into seven branches: cellular processes, environmental information processing, genetic information processing, metabolism, organicismal systems, and drug development. Further classification statistics are performed under each branch. The figure below (Figure 4B) shows the KEGG pathway annotation classification result for the selected gene set. Dozens of genes were assigned to the processes of the biosynthesis of secondary metabolites, starch and sucrose metabolism, carbon metabolism, flavonoid biosynthesis, and the biosynthesis of amino acids. Over 4000 genes were involved in global and overview maps, over 2000 genes were clustered in the carbohydrate metabolic process, and 689 genes were involved in secondary metabolite biosynthesis. 

These genes were further enriched with GO terms from three dimensions using an enriched bubble graph (Figure 4B). The size of the bubble indicates the number of genes annotated to a KEGG pathway, and several pathways were annotated: phenylpropanoid biosynthesis (392 genes); flavonoid biosynthesis (123); cyanoamino acid metabolism (123); indole alkaloid biosynthesis (91); stilbenoid, diarylheptanoid and gingerol biosynthesis (60); tropane, piperidine and pyridine alkaloid biosynthesis (56); phenylalanine metabolism (78); Isoquinoline alkaloid biosynthesis (54); and isoflavonoid biosynthesis (38). 

To further determine the biological functions of genes expressed during phenol synthesis, we mapped these genes to terms in the KEGG database. Among the mapped pathways, the biosynthesis of phenol compounds is involved in the pentose phosphate, shikimate, flavonoid and phenylpropanoid pathways in plants. In the synthesis of phenolic compounds, the first procedure is the commitment of glucose to the pentose phosphate pathway (PPP) and the transformation of glucose-6-phosphate irreversibly to ribulose-5-phosphate. PPP also produces erythrose-4-phosphate along with phosphoenolpyruvate from glycolysis, which is then used through the phenylpropanoid pathway to generate phenolic compounds after being channeled to the shikimic acid pathway to produce phenylalanine. Genes involved in the pentose phosphate, shikimate, flavonoid and phenylpropanoid pathways are shown in Figure 4C and Appendix A, where high-expressing genes were selected.

### 2.5. Key Enzymes Related to Phenols Synthesis during Rhizome Development

The committed enzymes analyzed in the transcriptome were dihydroxyacetone transferase, dihydroxyacetone synthase, formaldehyde transketolase (EC 2.7.1.2), glucose-6-phosphate isomerase (EC 5.3.1.9), phosphogluconate dehydrogenase (NADP+/–dependent, decarboxylating) (EC 1.1.1.44/42), ribose-5-phosphate isomerase (EC 5.3.1.6), dihydroxyacetone synthase (EC 2.2.1.2), and N-acetylglutamate synthetase (EC 2.3.1.1) in the pentose phosphate pathway and phenylalanine ammonia lyase (EC 4.3.1.24), tyrosine ammonia lyase (EC 4.3.1.23), cinnamate-4-hydroxylase (EC 1.14.13.11), coumaroyl:CoA ligase (EC 6.2.1.12), chalcone synthase (EC 2.3.1.74), chalcone isomerase (EC 5.5.1.6), flavanone-3-hydroxylase (EC 1.14.11.9), flavonol synthase (EC 1.14.11.23), dihydroflavonol 4-reductase (EC 1.1.1.219), and leucoanthocyanidin dioxygenase (EC1.14.11.19) in the phenylpropanoid biosynthesis pathway. As shown in Figure 5, two transcripts encoding glucokinase to catalyze the phosphorylation of glucose to produce glucose 6-phosphate were slightly expressed in S1. Four transcripts encode the phosphogluconate dehydrogenase: transcript 104597825 was highly expressed in all six samples, and transcript 104600406 specifically expressed in S2. Twenty homologs of ribose-5-phosphate isomerase and transcripts 104591772 and 104599199 were highly expressed in all samples; 104589021 and 104607784 showed completely opposite expression patterns between the first stage (S1 and S2) and the second stage (S3, S4, S5, and S6). Homologs of glucose-6-phosphate isomerase showed stable expression in all samples. In the phenylpropanoid biosynthesis pathway, 11 transcripts encoding phenylalanine ammonia lyase were identified during rhizome development; 104589393 was highly expressed in the S3 stage, 104601277 was highly expressed in S1, S2, and S3, and 104595050 was expressed in all these samples. Homologs of tyrosine ammonia lyase seemed to be slightly expressed in S1. Transcripts encoding chalcone synthase were upregulated in S1 and S3. We identified seven transcripts encoding flavanone-3-hydroxylase, while only three transcripts (104595860, 104605566 and 104605233) were expressed during rhizome development, and 104605233 was obviously upregulated in all these samples. Homologs of genes encoding flavonol synthase were analyzed, and the transcripts (104594902 and 104609672) were specifically expressed in S1 and S2 stages. Our results provide an overview of transcripts encoding the committed enzymes involved in phenols biosynthesis and clear clues to the study of it. 

### 2.6. Differentially Expressed Genes (DEGs) during Rhizome Development

There were 23,475 genes annotated. Further gene ontology (GO) enrichment was analyzed for each group, indicating that genes evolving at different rates can be biased in their functional categories, including biological processes, cellular components and molecular functions (Figure 6A). Differences in gene expression at six stages during rhizome development were examined, and DEGs were identified by pairwise comparisons of the six libraries (Figure 6B). Comparisons of the six samples identified 3441, 6386, 4481, 2875, and 1834 DEGs in pairs of S1 vs. S2, S2 vs. S3, S3 vs. S4, S4 vs. S5, and S5 vs. S6. The total number of DEGs across the three stages was much higher in S2 than S3 (Figure 6C). As shown in Figure 6C, 2609 DEGs were only detected in S2 vs. S3, 1396 DEGs in S3 vs. S4, and 770 DEGs in S1 vs. S2; 590 DEGs were uniquely detected in S4 vs. S5, and 213 DEGs in S5 vs. S6. A total of 143 DEGs were found in all samples. This indicated that more DEGs participated in the rhizome formation of S2 and S3 and the DEGs dropped off in the other samples. 

Transcriptional regulation revealed by RNA-seq was confirmed in a biologically independent experiment using qRT-qPCR. Ten differentially expressed genes involved in phenol biosynthesis were selected to design gene-specific primers (Appendix A), and a list of these gene annotations are shown in Figure 7B. The transcriptional levels of anthocyanidin reductase (104585744), 4-coumarate-CoA ligase-like (104587082), flavonoid hydroxylase (104590491), chalcone synthase (104591893), probable chalcone-flavonone isomerase 3 (104593770), anthocyanidin reductase-like (104593362), phenylalanine ammonia-lyse-like (104600622), phenylalanine ammonia-lyse G4 (104601277), leuconanthocyanidin reductase-like (104601230), and chalcone synthase (104602160), key enzymes in the secondary metabolites, are shown on a heatmap in Figure 7A. Further testing was done through qRT-PCR as shown in Figure 7C, and the data can be seen in Appendix A. Generally, the findings were consistent with RNA-seq results. As verified by qRT-PCR, anthocyanidin reductase (104585744), 4-coumarate-CoA ligase-like (104587082), flavonoid hydroxylase (104590491), and chalcone synthase (104591893) were downregulated in rhizome formation, and anthocyanidin reductase-like (104593362) and phenylalanine ammonia-lyse-like (104600622) were induced in rhizome formation. 

## 3. Discussion 

Phenolics are the most pronounced secondary metabolites found in plants, and their distribution is shown throughout the entire metabolic process. Synthesis of phenols is involved in several pathways, such as the pentose phosphate shikimate, flavonoid and phenylpropanoid pathways [16,17]. In recent years, many regulatory mechanisms have been carried out through transcriptome sequencing [6,23,24,25,26]. Many genes have been annotated and identified to be involved in different regulation mechanisms in these studies, deepening our comprehension even though there are still unknown genes. In this study, transcriptomic analyses of six samples during lotus rhizome formation were performed using a high-throughput tag sequencing technique. About 23 million high-quality reads were generated, and 92.14% of these were mapped to the reference genome. The samples were divided into two stages (S1, S2 and S3, S4, S5, S6), and we identified that 23,475 genes and 689 genes were involved in the biosynthesis of secondary metabolites. The differential rate of expression in the two stages suggests that the cell divisions changes from elongation to swelling. A complex genetic crosstalk-regulated network involved in the biosynthesis of phenolic compounds was found during the development of the lotus rhizome. The expression patterns of key enzymes assigned to the synthesis of phenolic compounds were also analyzed. Moreover, comparative transcriptomic analysis detected several differentially expressed genes required for phenolic compound biosynthesis, and as described above, 10 candidate genes involved in the synthesis of phenolic compounds were verified through qRT-PCR. This work lays a foundation for future studies on molecular mechanisms of phenolic compound biosynthesis during rhizome formation.

Previous studies have uncovered that fruits and vegetables such as grape, lotus, pomegranate, cranberry, and strawberry that are rich in antioxidant compounds reduce the risk of chronic diseases in humans [27,28,29]. Polyphenol-rich foods play important anti-cancer, anti-viral, and anti-oxidant roles, and have hypoglycemic, hypo-lipidemic, and anti-inflammatory activities to prevent the development of chronic diseases. Various analytical methods are used to separate and determine the phenolic compounds, including high-performance liquid chromatography (HPLC) and gas chromatography (GC) [30]. The contents of phenolic compounds in plants have been measured [31,32]. As reported in previous studies, the total phenolic content of flaxseed meal ranges from 0.355 to 0.442 g/100 g, and 26–29% of that comprises insoluble bound phenolic compounds [33,34]. In flaxseed, the content of phenolic compounds is 0.8–1 g/100 g, and esterified phenolic acids compose approximately half of the total. In soybean flake, the level of phenolic compounds ranges from 0.025 g/100 g, while in soybean flour, the amount is about 0.074 g/100 g [35,36]. In defatted rapeseed meal, full-fat rapeseed flour and canola meal, the total content of phenolic compounds is 1.84 g/100 g, 1.28 g/100 g and 0.1 g/100 g, respectively [32]. However, most studies of phenolic compounds in lotus rhizome have focused on browning in relation to phenolic metabolism during storage. Our study describes a complex genetic crosstalk-regulated network involved in the biosynthesis of phenolic compounds during lotus rhizome development and identifies the transcription levels of key enzymes in phenol biosynthesis. As phenylalanine ammonia lyase (PAL) is the first key enzyme involved in phenol biosynthesis in plants [37], 11 transcripts encoding PAL were identified in our study, of which 104589393, 104601277, and 104595050 were highly expressed in these samples. Transcripts encoding glucose-6-phosphate dehydrogenase (G6PDH), tyrosine ammonia lyase (TAL), cinnamate-4-hydroxylase (C4H), coumaroyl, CoA ligase (4CL), chalcone synthase (CHS), chalcone isomerase (CHI), flavanone-3-hydroxylase (F3H), and flavonol synthase (FLS) involved in phenols synthesis, were also identified in this study. Many genes involved in phenol biosynthesis pathways were proposed. These results may prove useful for a better understanding of the genetic mechanisms underlying phenol biosynthesis during lotus rhizome formation.

In *Nelumbo nucifera*, the regulatory gene circuitry of the rhizome formation process has been elucidated based on comprehensive transcriptome profiling [6,9,23], including the genome-wide regulation of lotus genes in rhizome development and in response to different stresses by high-throughput mRNA sequencing. These studies provide useful information on the molecular mechanisms underlying lotus formation and responses to different stresses, while our study confirms a higher expression of genes and provides an overview of the transcriptional changes involving the phenol biosynthesis pathways. We identified 23,475 genes, and 689 genes were involved in the biosynthesis of secondary metabolites. Comparisons of the six samples identified 3441, 6386, 4481, 2875, and1834 DEGs in pairs of S1 vs. S2, S2 vs. S3, S3 vs. S4, S4 vs. S5, and S5 vs. S6. Most of the DEGs were related to cellular processes, membrane parts, metabolic processes, and catalytic activity. We selected 10 DEGs involved in secondary metabolism and confirmed by qRT-PCR, and the results showed high correlation with RNA-seq data. However, the exact mechanisms of phenolic compound synthesis have not been elucidated and may be complex. This will be further investigated in future studies.

This study covers an analysis of the comprehensive transcriptomic data to provide a perspective on transcriptional changes and metabolic interconversions of phenols during rhizome development. Further analysis on the mechanisms of phenolic compound synthesis in lotus will provide insight and enable a deeper understanding of genetic involvement in the improvement of phenolic phytochemical, nutritional and health-relevant functional values.

## 4. Materials and Methods

### 4.1. Plant Materials and Determination of Total Phenolic Content

Elian No. 5, a widely cultivated species in China, was sampled after rhizome formation every week. Samples were collected in a lotus rhizome experimental field in Wuhan (N30°32′44.02″, E114°24′52.18″) in 2018. For the analysis of tag-sequencing and gene expression, rhizomes at various developmental stages were frozen at 4 °C to be cleaned, cut–up, and stored in liquid nitrogen for immediate RNA extraction. Total phenolic content was measured according to the Folin–Ciocalteu method [38].

### 4.2. RNA Isolation and Sequencing

Total RNA was extracted using the RNA reagent (OminiPlant RNA Kit, CWBIO, Beijing, China) and RNase-free DNaseI (Thermo, Shanghai, China) to remove genomic DNA contamination. RNA integrity was evaluated, with an RNA integrity number (RIN) above 6.5 for all samples. More than 3 μg of total RNA was used as input material for each RNA sample. Total RNA was treated by mRNA enrichment and rRNA removal: the enrichment of mRNA was performed with polyA tail using magnetic beads with OligodT, the hybridization of rRNA with a DNA probe, and the DNA probe was digested with DNaseI; then, the desired RNA was obtained with a purifying step. The obtained RNA was synthesized to form double-stranded DNA with the random N6 primer. The data obtained by sequencing are called raw reads or raw data, and then quality reads (QC) are performed on the raw reads to determine whether the sequencing data are suitable for subsequent analysis. Then, clean reads after filtering are compared to the reference genome. After the comparison, the statistical comparison ratio, the distribution of reads on the reference sequence, etc. are determined to find out whether the comparison result passes the QC of alignment. The quantitative analysis of genes is based on gene expression levels (principal components, correlation, differential gene screening, etc.), and significant enrichment analysis of GO functions on the differentially expressed genes was performed between the selected samples. Pathway-significant enrichment analysis, clustering, protein interaction networks and more in-depth mining analysis were performed.

Data processing and sequenced reads containing adapters, poly N or low-quality sequences (Q < 20) were removed, and the remainder were termed as clean reads. All the downstream analyses were based on high-quality clean data. The data were mapped to the reference genome of *Nelumbo nucifera* (China lotus 1.1) using TopHat2 v2.0.9 software.

### 4.3. Analysis and Mapping of RNA-Seq Reads

Raw data in the Fastq format were initially processed through in-house perl scripts. All downstream analyses were based on high-quality, clean data. Reference genome and gene model annotation files were downloaded directly from the genome website (http://www.ncbi.nlm.nih.gov/genome/genomes/14095). An index of the reference genome was built using Bowtie v2.0.6, and single-end clean reads were aligned to the reference genome using TopHat v2.0.9.

### 4.4. Differential Expression Analysis

HTSeq v0.5.4p3 was employed to count read numbers mapped to each gene. Reads per kilobase per million mapped reads (RPKM) of individual genes were calculated based on gene length and read counts mapped to each gene. Differential expression analysis was performed using the DESeq R package (1.10.1). The resulting *p*-values were adjusted using Benjamini and Hochberg’s approach to control the false discovery rate. Genes with an adjusted *p*-value of < 0.05 determined by DESeq were assigned as differentially expressed.

### 4.5. Gene Ontology and KEGG Ortholog Enrichment Analysis

Based on Wallenius non-central hyper-geometric distribution [39], the GO enrichment analysis of the differentially expressed genes (DEGs) was implemented by the GOseq R package 1.10.0, which can be adjusted for gene length bias in DEGs. KEGG Orthology-Based Annotation System (KOBAS) software was used to test the statistical enrichment of DEGs in KEGG pathways (FDR ≤ 0.05), which is also important for the development and utilization of lotus phenolic substances.

### 4.6. Quantitative Real-Time PCR Validation of RNA-Seq Data

A total of 10 DEGs involved in the synthesis process of secondary metabolites were further identified using qRT-PCR. The GenBank number and IDs of the DEGs are as follows: 104585744 (XM_010242720.2), 104587082 (XM_010244534.2), 104590491 (XM_010249177.2), 104591893 (XM_010251015.1), 104593362 (XM_010253129.2), 104593770 (XM_010253759.1), 104600622 (XM_010263680.2), 104601230 (XM_010264501.2), 104601277 (XM_010264565.2), 104602160 (NM_001318155.1). Relative gene expressions were normalized by comparison with the expression of β-actin (XM_010243420.2) and analyzed using the 2^–ΔΔ*C*T^ method. The sequences of oligonucleotide primers are listed in Appendix A. The data are shown as mean ± standard deviation (SD).

## Figures and Tables

**Figure 1 ijms-20-02735-f001:**
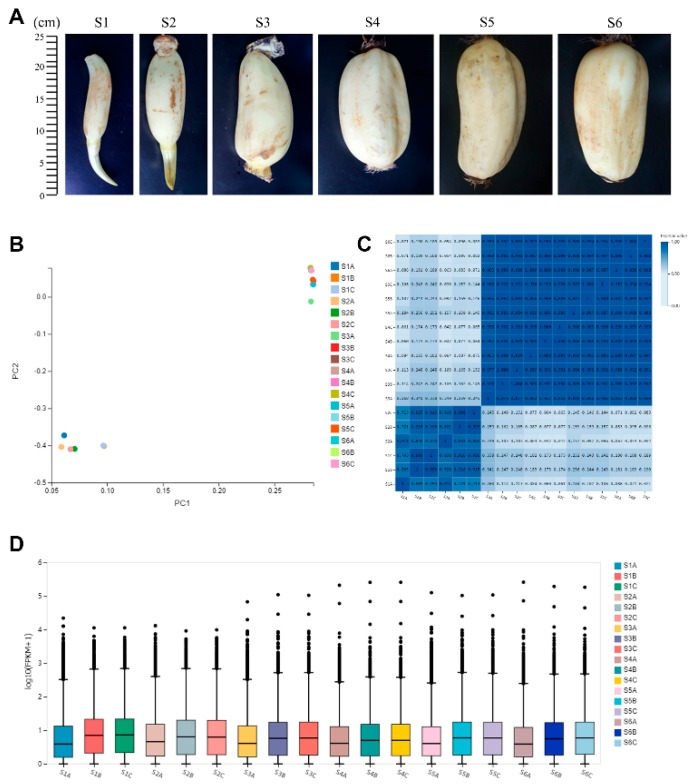
Transcriptome during rhizome development. (**A**) Six developmental stages of lotus rhizome growth used for gene expression in this study. (**B**) Principal component analysis of six samples. (**C**) Pearson correlation coefficient of gene expression levels between samples. (**D**) Distribution of gene expression levels in each sample.

**Figure 2 ijms-20-02735-f002:**
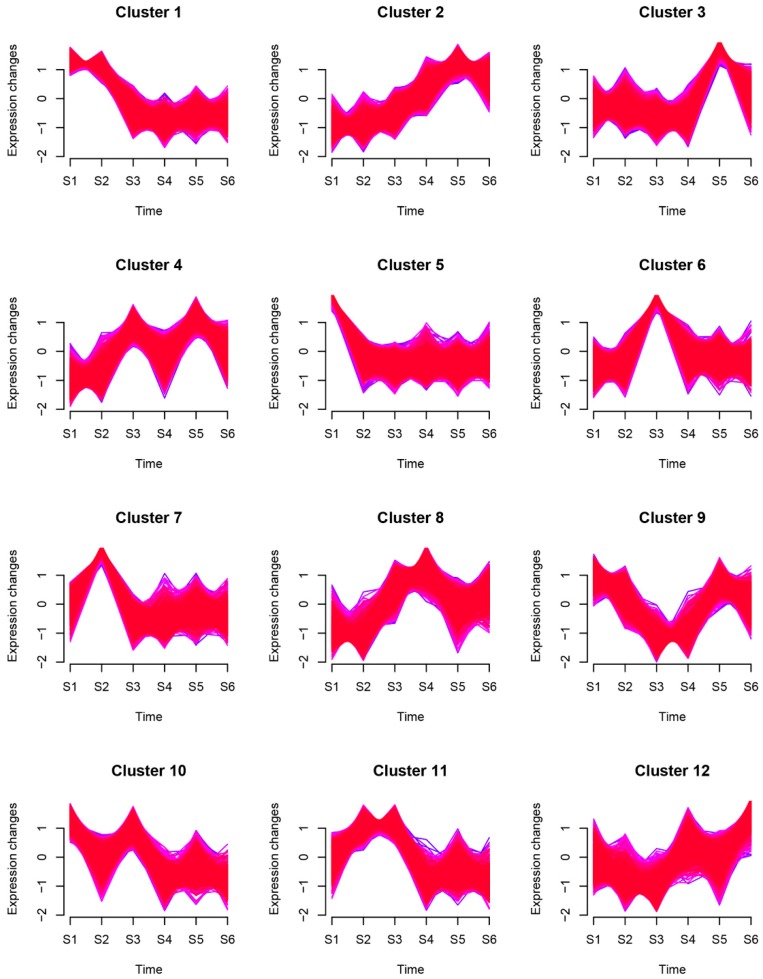
Time-associated gene clusters. According to the gene expression information, genes with similar expression patterns at different time points were clustered into different groups.

**Figure 3 ijms-20-02735-f003:**
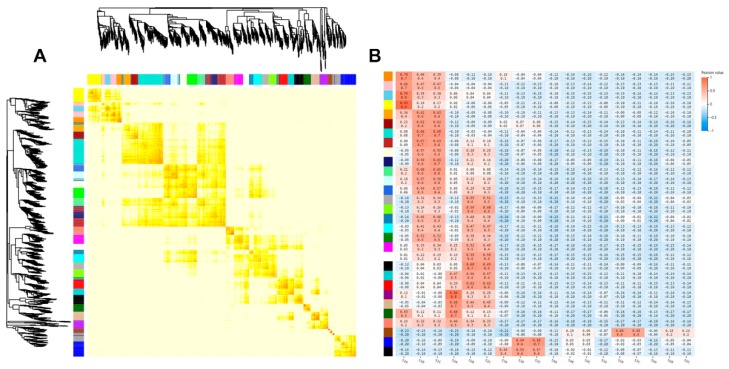
Co-expression network during rhizome development. (**A**) Weighted gene co-expression network analysis (WGCNA) analysis of RNA sequencing (RNA-seq) data from different stages of rhizome development. (**B**) Network hubs regulating genes in the differentiation stage. Colors indicate the genes displaying peak expression in corresponding stages.

**Figure 4 ijms-20-02735-f004:**
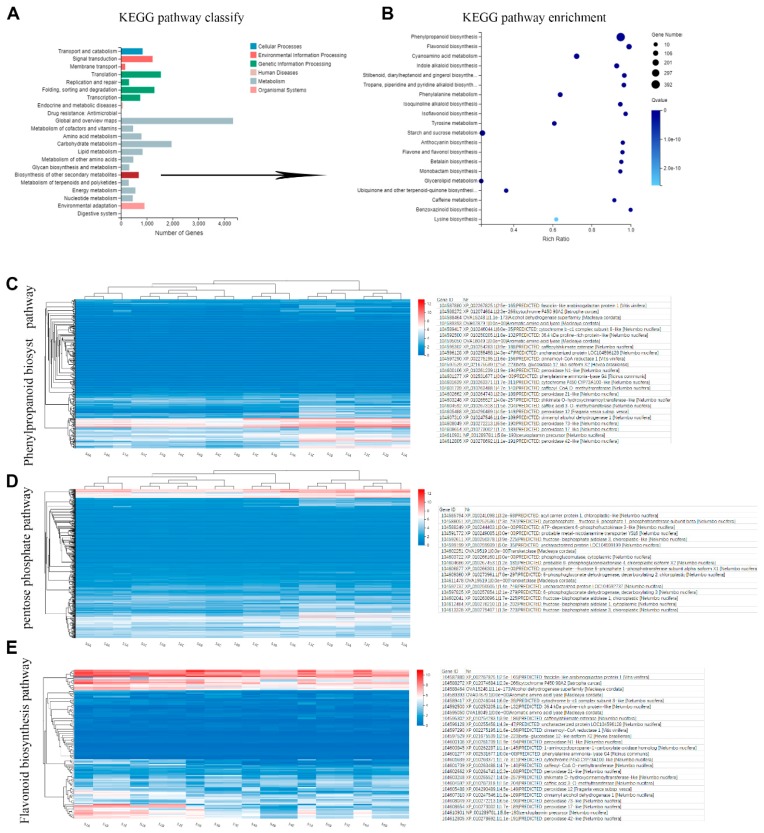
Overview of genes involved in biosynthesis pathway of phenols in six rhizome transcriptomes. (**A**) Gene ontology (GO) annotation of total genes in these transcriptomes. (**B**) Kyoto Encyclopedia of Genes and Genomes (KEGG) pathway enrichment of genes involved in secondary metabolism. Pathways were enriched with Q values. (**C**) Heatmap of expressed genes assigned to the phenylpropanoid biosynthesis pathway; highly expressed genes were selected and are listed on the right. (**D**) Heatmap of genes involved in the pentose phosphate pathway; highly expressed genes were selected and are listed on the right. (**E**) Heatmap of genes involved in the flavonoid biosynthesis pathway; highly expressed genes were selected and are listed on the right.

**Figure 5 ijms-20-02735-f005:**
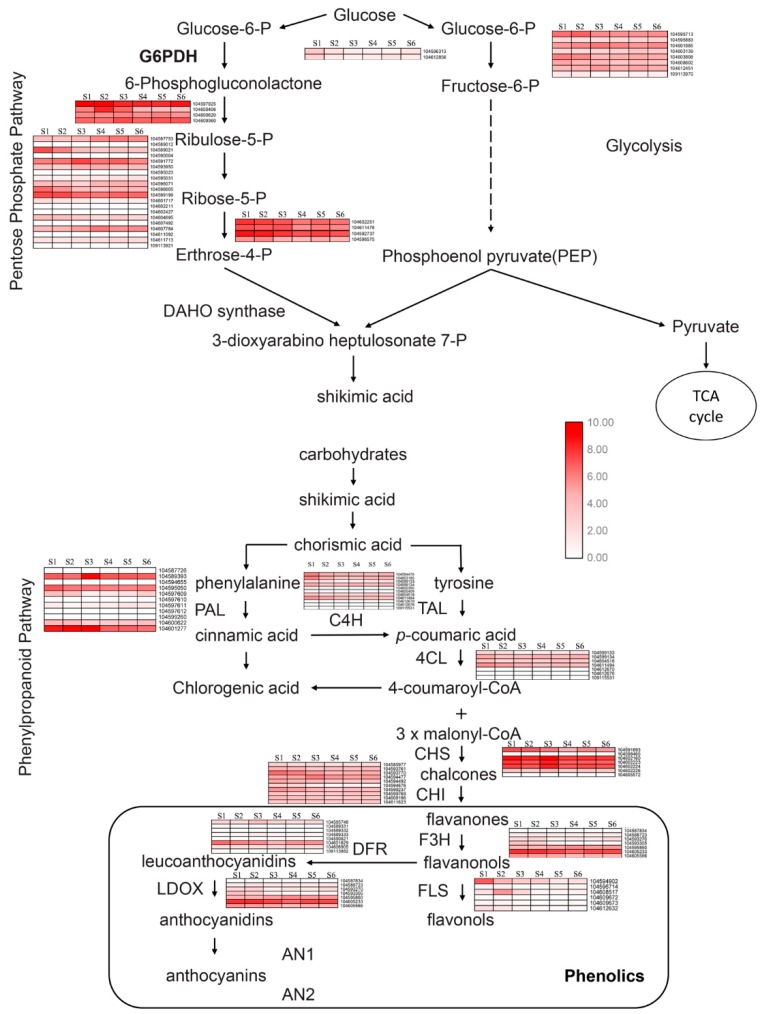
Expression patterns of expressed genes assigned to phenol biosynthesis in the S1, S2, S3, S4, S5, and S6 stage rhizome transcriptomes. Log-transformed expression values range from 0 to 10. Red gradations indicate up and downregulated transcripts.

**Figure 6 ijms-20-02735-f006:**
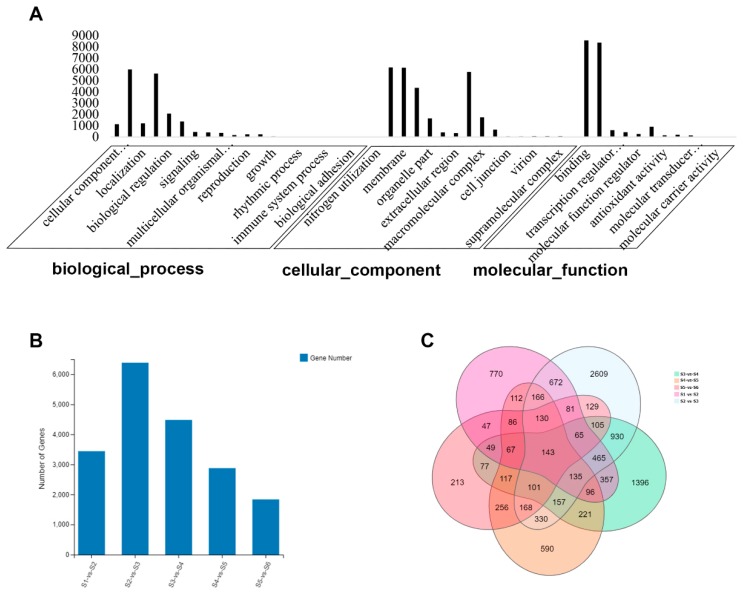
Differentially expressed genes (DEGs) during rhizome development. (**A**) GO annotation of total genes in the transcriptomes. (**B**) Histogram of DEGs at six stages. (**C**) Venn diagrams of DEGs among samples.

**Figure 7 ijms-20-02735-f007:**
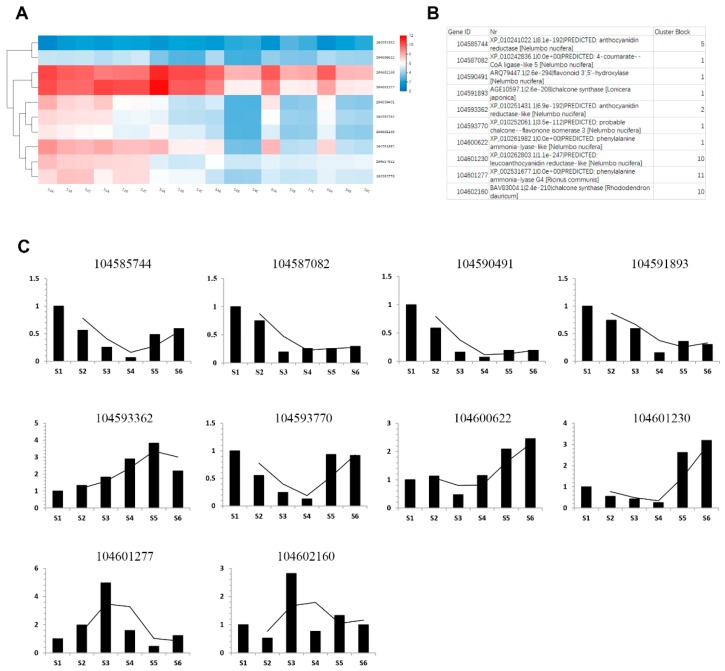
qRT-PCR confrmation of 10 candidate DEGs. (**A**) Heatmap of differentially expressed genes selected from the six rhizome transcriptomes. Log-transformed expression values range from 0 to 12. Red and blue indicate different transcript levels. (**B**) Detailed information of candidate genes. (**C**) Relative gene expression levels determined by qRT-PCR. Relative gene expressions were normalized by comparison with the expression of lotus β-actin and analyzed using the 2^−ΔΔ*C*T^ method. The expression values were adjusted by setting the expression of S1 to 1 for each gene. qRT-qPCRs for each gene used three biological replicates, with three technical replicates per experiment; error bars indicate standard error (SE).

**Table 1 ijms-20-02735-t001:** Summary statistics of clean reads in the rhizome transcriptomes of lotus.

Sample	Total Raw Reads (M)	Total Clean Reads (M)	Total Clean Bases (Gb)	Clean Reads Q20 (%)	Clean Reads Q30 (%)	Clean Reads Ratio (%)	Total Mapping (%)	Uniquely Mapping (%)
S1A	25.94	25.54	1.28	98.32	91.52	98.47	88.28	76.82
S1B	21.94	21.66	1.08	98.32	90.64	98.73	88.71	77.37
S1C	21.94	21.77	1.09	98.47	91.5	99.2	88.42	77.16
S2A	25.74	25.5	1.27	98.4	91.55	99.05	86.9	75.51
S2B	21.94	21.81	1.09	98.32	90.71	99.38	88.41	76.93
S2C	21.94	21.82	1.09	98.31	90.71	99.42	88.73	77.04
S3A	25.75	25.37	1.27	98.36	91.51	98.49	75.22	64.29
S3B	21.94	21.78	1.09	98.3	90.54	99.25	86.11	73
S3C	21.94	21.83	1.09	98.39	90.97	99.47	85.73	72.65
S4A	26.18	24.85	1.24	95.33	83.71	94.93	83.75	67.79
S4B	21.94	21.77	1.09	98.4	90.93	99.22	86.99	69.13
S4C	21.94	21.84	1.09	98.34	90.6	99.53	88.11	70.06
S5A	28.4	25.62	1.28	96.62	86.76	90.2	80.08	66.9
S5B	21.94	21.8	1.09	98.42	91.03	99.37	85.09	71.41
S5C	21.94	21.85	1.09	98.55	91.5	99.58	85.77	71.73
S6A	25.31	24.44	1.22	98.15	90.73	96.57	83.23	65.4
S6B	21.94	21.75	1.09	98.4	91.03	99.13	87.41	71.06
S6C	21.94	21.82	1.09	98.38	90.82	99.44	87.32	71.43

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
