# Peer review of "Transcription Profiles Reveal the Regulatory Synthesis of Phenols during the Development of Lotus Rhizome (Nelumbo nucifera Gaertn)"

_ijms, 2019, doi:10.3390/ijms20112735_

Reviewer 1 Report

The manuscript entitled"Transcription profiles reveal the regulatory synthesis of phenols during the development of lotus root" identified genes in phenylpropanoid biosynthesis pathway, in pentose phosphate pathway and in flavonoid biosynthesis. Transcriptome analysis, including PCA, gene clusters, WGCNA co-expression analysis, GO and validation of gene expression by RT-PCR, is well-performed. The bioinformatic analysis provides valuable information for molecular mechanisms underling phenolic compound biosynthesis in lotus. However, few concerns need to be addressed before acceptance of publication.

1. The samples collected in this study is rhizome (stem) instead of root. Please remove the term "root" to avoid confusion.

2. It is highly recommend that the author analyze the phenolic profiles and antioxidant activity to explain the difference between samples of  two stage (S1, S2 and S3, S4, S5, S6). Please refere to the reference:

Phenolic Profiles and Antioxidant Activity of Lotus Root Varieties June 2016Molecules 21(7):863

DOI: 10.3390/molecules21070863

3. According to the study, which stage is better (more beneficial phenolic compounds) for collection in terms of agriculture?

4. Editing of English is required.

Author Response

Point 1: The samples collected in this study is rhizome (stem) instead of root. Please remove the term "root" to avoid confusion.

Response 1: Thank you for your suggestions and comments. We have revised it.

Point 2: It is highly recommend that the author analyze the phenolic profiles and antioxidant activity to explain the difference between samples of  two stage (S1, S2 and S3, S4, S5, S6). Please refere to the reference:

Phenolic Profiles and Antioxidant Activity of Lotus Root Varieties June 2016 Molecules 21(7):863

DOI: 10.3390/molecules21070863

Response 2: Thank you for your suggestion. We have tested the phenolic profiles in different samples and add these data as the supplementary figure 1. The reference was added in the manuscript.

Point 3: According to the study, which stage is better (more beneficial phenolic compounds) for collection in terms of agriculture?

Response 3: As the supplementary figure 1 showed, the second stage is better for collection of phenolic compounds.

Point 4:  Editing of English is required.

Response 4: We have asked for the English editing service of MDPI in the improvement of this manuscript.

Reviewer 2 Report

In a manuscript authors present comparative analysis of gene expression over the Lotus root development, with the special focus on phenols biosynthesis pathway genes.  It will be worth to mention if  the content of phenols during the development of roots is known. If yes, correlation of physiological parameters with the gene expression would increase the value of the study and help to draw conclusions. In Results and Discussion parts, the main founding regarding to expression of genes that are involved in phenols biosynthesis should emphasized. The first impression is that only slight differences for a few genes are observed over time.

The text is written carelessly (for example probably a fragment of a sentence in lines 234-235 or 467-468 ), which makes difficult to follow the manuscript.

In the Results section:

1.       Range for uniquely mapped reads provided in line 123 does not match the Table1. 

2.       Line 146-148 contains a legend to the figure 1C, I would suggest to transfer it to the description of Figure 1 . Similarly, information regarding Figure 1D (lines 152-153) should be part of Figure 1 description.

3.       The comment to the distribution of gene expression (lines 153-155) is not clear. There is no statistical support for statement that 'gene expression levels are higher'. I agree that some outliers present in samples S3-S6 have  higher expression values. The valuable would be information if outliers shown in Figure 1D represent  the same gens for each sample.

4.       In line 169 'the cluster 1, 6, 7 significantly expressed in the S5, S3, and S2' refers to clusters 3, 6, and

5.       The  meaning of the sentence in lines 169-170 is difficult to understand.

6.       Line 206: 'Venn diagram of DEGs' while there is no such diagram in Figure 4.

7.       I suggest to put the key enzymes involved in phenols synthesis mentioned in text (lines 248257)  in a table.

8.       Statement 'comparisons of the six libraries (line 291) suggests that  six libraries were prepared and each of them was sequenced 3x, giving technical replicates.

9.       Order of genes in Figure7 A is different from the order in B and C. It is confusing, especially if quality of Figure does not allow to read label.  Figure 7A does not show genes, that are induced in the rhizome formation.

10.   In each section long description of analytical method is included, while results are only briefly described.

In the Materials and Methods section:

1.       In section 'RNA isolation and sequencing' part between lines 424-440 should be removed. It is description of methods explained in later in following sections.

Author Response

Point 1: The text is written carelessly (for example probably a fragment of a sentence in lines 234-235 or 467-468 ), which makes difficult to follow the manuscript.

Response 1: Thank you for your suggestion. We have tested the phenolic profiles in different samples and add these data as the supplementary figure 1. The sentences were revised and we have asked for the English editing service of MDPI in the improvement of this manuscript.

Point 2: In the Results section:

(1). Range for uniquely mapped reads provided in line 123 does not

match the Table1. 

Response (1): The mistake has been corrected.

(2). Line 146-148 contains a legend to the figure 1C, I would suggest to transfer it to the description of Figure 1 . Similarly, information regarding Figure 1D (lines 152-153) should be part of Figure 1 description.

Response (2): We agree with the reviewer's comment and we have revised it.

(3). The comment to the distribution of gene expression (lines 153-155) is not clear. There is no statistical support for statement that 'gene expression levels are higher'. I agree that some outliers present in samples S3-S6 have higher expression values. The valuable would be information if outliers shown in Figure 1D represent the same gens for each sample.

Response (3): Thank you for your suggestion. The box plot diagram for each region corresponds to five statistics (upper to lower, upper quartile, median, lower quartile, lower limit, where the upper and lower limits do not take into account outliers). And we have rephase the sentence.

(4). In line 169 'the cluster 1, 6, 7 significantly expressed in the S5, S3, and S2' refers to clusters 3, 6, and

Response (4): Thank you for your suggestion. We have rephase the sentence.

(5). The  meaning of the sentence in lines 169-170 is difficult to understand.

Response (5): Thank you for your suggestion. We have rephased the sentence.

(6). Line 206: 'Venn diagram of DEGs' while there is no such diagram in Figure 4.

Response (6): The mistake has been corrected.

(7). I suggest to put the key enzymes involved in phenols synthesis mentioned in text (lines 248257)  in a table.

Response (7): Thank you for your suggestion. These data are also shown in supplementary table 1 and we have rephased the sentence.

(8). Statement 'comparisons of the six libraries (line 291) suggests that  six libraries were prepared and each of them was sequenced 3x, giving technical replicates.

Response (8): Thank you for your suggestion. These data are also shown in supplementary table 3 and we have rephased the sentence.

(9). Order of genes in Figure7 A is different from the order in B and C. It is confusing, especially if quality of Figure does not allow to read label.  Figure 7A does not show genes, that are induced in the rhizome formation.

Response (9): Thank you for your suggestion. These genes were selected out as the different expressed genes in the secondary metabolism through the previous study and our results. Another aim is to verify out results.

(10). In each section long description of analytical method is included, while results are only briefly described.

Response (10):  Response: Thank you for your suggestion.

Point 3: In the Materials and Methods section:

In section 'RNA isolation and sequencing' part between lines

424-440 should be removed. It is description of methods explained in later in following sections.

Response 3:Thank you for your suggestion. We have revised it.

Reviewer 3 Report

The MS is badly written, English must be completely revised, and data are insignificant. Authors present / describe a huge number of genes without focusing on something in particular so that it is not possible to understand the purpose of the work.  The conclusion is simply a generic “This work lay a foundation for future studies on molecular mechanisms of phenolic compounds biosynthesis during rhizome formation”

Therefore, It deserves to be rejected.

Moreover:

Authors confuse root with rhizome;

Authors indicate two stage of developments with 6 labels (S1-S6), that doesn’t work. At line 55-56 they describe 4 different development stages; Line 112: “Six samples”, line 115-116: “six libraries, three biological replicates each sample”, Legend Figure 1: “six developmental stages”;  line 343: “The samples are divided into two stage (S1, S2 and S3, S4, S5, S6)” - confusing and contradictory statements that makes the data comparisons presented in the MS incomprehensible;

Specie names (mainly in References) are not in Italics.

Author Response

Point 1: The MS is badly written, English must be completely revised, and data are insignificant. Authors present / describe a huge number of genes without focusing on something in particular so that it is not possible to understand the purpose of the work.  The conclusion is simply a generic “This work lay a foundation for future studies on molecular mechanisms of phenolic compounds biosynthesis during rhizome formation”

Therefore, It deserves to be rejected.

Response 1: Thank you for your suggestion. We also tested the phenolic profiles in different samples and add these data as the supplementary figure 1. This study is the first step of finding out the genes we interested in the formation of the rhizome and we have achieved this aim, there would be a lot of work to do based on this study. Moreover, we have asked for the English editing service of MDPI in the improvement of this manuscript.

Point 2: Moreover, Authors confuse root with rhizome;

Response 2: The mistake has been corrected.

Point 3: Authors indicate two stage of developments with 6 labels (S1-S6), that doesn’t work. At line 55-56 they describe 4 different development stages; Line 112: “Six samples”, line 115-116: “six libraries, three biological replicates each sample”, Legend Figure 1: “six developmental stages”;  line 343: “The samples are divided into two stage (S1, S2 and S3, S4, S5, S6)” - confusing and contradictory statements that makes the data comparisons presented in the MS incomprehensible;

Specie names (mainly in References) are not in Italics.

Response 3: Thank you for your suggestion. The mistake has been corrected.
